

marina.levy@locean.ipsl.fr

# Satellite data reveal earlier and stronger phytoplankton blooms over fronts in the Gulf Stream region

Clément Haëck[1], Marina Lévy[1], Inès Mangolte[1], and Laurent Bopp[2]

[1]LOCEAN-IPSL, Sorbonne Université, CNRS, IRD, MNHN, Paris, France
[2]LMD-IPSL, École Normale Supérieure / Université PSL, CNRS, École Polytechnique, Paris, France
**Correspondence:** M. Lévy

**Abstract.** Fronts affect phytoplankton growth and phenology by locally reducing stratification and increasing vertical nutrient supply. Biomass peaks at fronts have been observed in-situ and linked to local nutrient upwelling, and reduced stratification over fronts has been shown to induce earlier blooms in numerical models. However observation of these biophysical interactions

through satellite imagery have been scarce, despite the opportunity to quantify them at synoptic scales. Here we used twenty years of SST and Chlorophyll-*a* satellite data in a large region surrounding the Gulf Stream to quantify the impact of fronts on phytoplankton in contrasting regimes, from oligotrophy to bloom, and throughout the year. We computed an heterogeneity index HI from SST, and used it to sort fronts into weak and strong fronts based on HI thresholds. We observed that the localization of strong fronts corresponded to western boundary current fronts, and weak fronts to more ephemeral submesoscale

fronts. We compared Chlorophyll-*a* distributions over strong fronts, weak fronts and outside of fronts. We assessed three metrics, the local enhancement of Chlorophyll-*a* over fronts, the global enhancement of Chlorophyll-*a* due to fronts at the scale of the region, and the lag in spring bloom onset due to fronts. We found that weak fronts lead to a local enhancement of Chlorophyll-*a* weaker than strong fronts, but because they are also more frequent they contribute equally to the regional Chlorophyll-*a* budget. We also find the the local enhancement of Chlorophyll-*a* was two to three times larger for the spring

bloom than in the oligotrophic subtropical gyre. We also provide observational evidence that blooms start earlier over fronts, by one to two weeks. Nevertheless our results suggest that the spectacular impact of fronts at the local scale may be misleading, considering their impact on a regional scale budget remains limited.

**Plain Language Summary**

Phytoplankton vary in abundance in the ocean over of large regions, and with the seasons, but also because of small-scale het-

erogeneities in surface temperature, called fronts, where phytoplankton growth can be favoured. Here, using satellite imagery, we found that fronts enhance phytoplankton much more where it is already growing well, but despite large local increases the enhancement for the region is modest (5%). We also found that blooms start by one to two weeks earlier over fronts.



## 1 Introduction

Phytoplankton form the basis of marine food webs and are key players in the ocean carbon cycle. The transport of limiting
nutrients to the sunlit euphotic layer by advective and convective processes, and the amount of light received by the cells —
which is closely related to the stratification of the water column — are two important factors that control their growth. As
there are marked contrasts in nutrient and light availability in the ocean, it follows that the global ocean can be divided into
different regional biomes (or bioregions), characterized by different phytoplankton abundances and seasonality (Longhurst,
2007; Vichi et al., 2011; Bock et al., 2022). The contrasts between biomes are largely explained by consistent physical forcings
and environmental conditions, operating at the biome scale, which determine how the two main controlling factors, nutrient
and light, limit growth. For example, subtropical gyres are areas where wind-driven circulations induce a deepening of the
thermocline and nutricline, resulting in oligotrophic biomes where productivity is relatively constant and low throughout the
year; at higher latitudes, the nutricline is shallower and the strong seasonality of the vertical mixing will induce a multi-stage
operation, with a time of reduced productivity and convective nutrient supply in winter when the mixing is strong, and a bloom
in spring when the stratification sets in (Wilson, 2005; Williams and Follows, 2011).

In addition to these large-scale patterns, there has been considerable evidence over past years that submesoscale motions
actively influence the nutrient and light environments at the horizontal scale of the order of 1–50km (see reviews by Lévy
et al., 2012; Mahadevan, 2016; Lévy et al., 2018). Submesoscale motions arise dynamically through advective interactions
involving mesoscale strain that continuously create sharp submesoscale density fronts, or at more steady wind-driven density
fronts, such as western boundary currents (Thomas et al., 2008; McWilliams, 2016; Mahadevan et al., 2020). These fronts are
characterized by an energetic secondary vertical circulation, with upwelling on the warm side of the front, and downwelling on
the cold side. Submesoscale dynamics also involve restratification and suppression of vertical mixing at the front (Thomas and
Ferrari, 2008). Thus submesoscale dynamics occurring at ocean fronts may affect phytoplankton in various ways. Of interest
here, the upward branch of the secondary circulation may enhance phytoplankton growth by transporting nutrients into the
euphotic zone, while the downward branch may subduct biomass and excess nutrients into the subsurface (Calil et al., 2011;
Omand et al., 2015; Hauschildt et al., 2021). In addition, in highly seasonal regimes where productivity is slowed in winter
due to deep mixing, submesoscale restratification may promote localized phytoplankton blooms before the large-scale outburst
associated with seasonal stratification (Mahadevan et al., 2012).

Despite numerous local observations and a strong theoretical basis for these processes (e.g. recent studies by Marrec et al.,
2018; Little et al., 2018; Verneil et al., 2019; Ruiz et al., 2019; Uchida et al., 2020; Kessouri et al., 2020; Tzortzis et al.,
2021), their integrated contribution at the scale of regional biomes is still largely unknown. Indeed, as most fronts form, move
and dissipate continuously on time scales of days to weeks, they are particularly difficult to sample, and this limitation is
reinforced by the fact that only a limited number of fronts can be observed with in situ field observations. Thus satellite-
derived estimates of Chlorophyll-*a* (hereafter Chl-*a*) although limited to the surface and an imperfect proxy for phytoplankton
biomass, are the only data that allow to track the impact of fronts synoptically over large areas. A first attempt to assess the
contribution of fine scales to regional satellite Chl-*a* budgets was based on a geostatistical analysis derived from data at 9km



resolution (Doney et al., 2003), extended later in Glover et al. (2018), with which they examined the change in spatial variance with distance. This methodology was too coarse to reveal the impact of submesoscales but confirmed the important role of mesoscale eddies in stirring large scale gradients of phytoplankton abundance. The role of submesoscales has been assessed

with three different methods. Guo et al. (2019) combined ocean color data with altimetry and drifting floats, and estimated that, over subtropical gyres of the global ocean, the respective contributions of mesoscales and submesoscales to high Chl-*a* anomalies were comparable in magnitude. Keerthi et al. (2022) proposed an approach based on deconvolution of local Chl-*a* time series into different timescales; they observed that sub-seasonal time scales contributed roughly 30% of the total satellite Chl-*a* variance and were associated with small (< 100km) spacial scales. Finally, the most quantitative approach was proposed

by Liu and Levine (2016), which they applied to the North Pacific Subtropical Gyre. They detected sea-surface temperature (SST) fronts by computing an index that measures the local heterogeneity of the SST field from satellite SST data. This allowed them to compare satellite Chl-*a* values over areas impacted by fronts (characterized by a large value of the heterogeneity index) with values over areas that were not impacted. They found that the increase in Chl-*a* over the fronts was negligible in summer but reached almost 40% in winter.

Here we built on this last approach, and quantify the excess Chl-*a* due to the presence of fronts at the scale of biomes. This more global quantification will depend on several factors, since the local contribution of fronts to Chl-*a* depends on many factors. First it depends on how efficient fronts are at supplying nutrients, which itself depends on how deep the fronts reach into the nutricline, and of the seasonality of this supply. It should be noted that the overall efficiency of fronts has been questioned (Lévy et al., 2018); firstly because submesoscale fronts are more numerous in winter when convective nutrients input is also

greatest; and secondly because the submesoscale vertical currents are often trapped within the mixed-layer, and may not reach the nutricline, which is often well below the base of the boundary mixed-layer. Second, the overall contribution of fronts will differ between biomes, with submesoscale vertical advection of nutrients likely to be more important in oligotrophic biomes where other nutrient supply routes are scarce, and submesoscale restratification in blooming biomes. Finally, the contribution of fronts will depend on their spatio-temporal footprint, which also varies seasonally (Callies et al., 2015) and regionally

(Mauzole, 2022).

Thus, more precisely, our intention is to explore and quantify how the contribution of fronts to biome-scale Chl-*a* varies in three contrasted biomes, ranging from subtropical to subpolar, varies along the year, and varies with the occurrence and strength of fronts. We focus our analysis on the North-Atlantic region surrounding the Gulf Stream, where multiple biomes and fronts of different strengths are found in a limited geographical area (Bock et al., 2022) with strong seasonality. In the south,

our study area encompasses part of the North Atlantic subtropical gyre, characterized by an oligotrophic regime, year-long low productivity. In the north, north of the Gulf Stream jet, is a more productive subpolar regime characterized by a recurrent spring bloom. In between, there is a moderately productive regime, with maximum productivity in winter. Another feature that makes this study area particularly relevant is that it has different types of fronts. On the one hand, there are two strong persistent fronts, the Gulf Stream and the shelf-break front, which are held in place by topography and atmospheric circulation,

and which are both associated with strong and deep-reaching vertical circulations (Flagg et al., 2006; Liao et al., 2022). But there are also more ephemeral and weaker fronts, created by mesoscale strain that are continuously forming at more random



locations (Drushka et al., 2019; Sanchez-Rios et al., 2020). We use satellite data of Chl-*a* and SST, and extend on the approach of Liu and Levine (2016), to distinguish between persistent and ephemeral fronts. We evaluate the impact of both types of fronts on Chl-*a* on the basis of three indicators, the excess (or deficit) Chl-*a* over fronts at the local scale of the front, the

surplus Chl-*a* attributable to fronts at the scale of regional biomes, and the change in the timing of the Chl-*a* spring bloom over fronts.

## 2   Methods and data

Our approach combines daily satellite SST data, which are used to detect fronts and sort them by their strength, with daily satellite surface Chl-*a*, from which we derive anomalies over fronts. Our region of interest is the North Atlantic from 15°N to

55°N, and from 40°W to the North American shelf break (Fig. 1). This region covers three biomes, a more oligotrophic one in the south, a more productive one in the north where a spring bloom occurs, and an intermediate biome between the two, described below. We will refer to them in the following as permanent subtropical biome (PSB), seasonal subtropical biome (SSB) and subpolar biome (PB), moving from south to north. They are separated from south to north by a fixed boundary at 32°N and by a variable boundary near 40°N dynamically set at each time step to the instantaneous position of the Gulf Stream.

### 2.1   Data

For Chl-*a*, we used the L3 product distributed by ACRI-ST over the period 2000–2020, generated by Copernicus-GlobColour, constructed with data from different sensors (SeaWIFS, MODIS Aqua & Terra, MERIS, VIIRS-SNPP & JPSS1, OLCI-S3A & S3B) merged and reprocessed, available daily at 4km resolution (Maritorena et al., 2021).

    For SST, we used the European Space Agency Sea Surface Temperature Climate Change Initiative analysis product version

2.1 (Merchant et al., 2019; Good et al., 2020a, b), also available daily at 4km resolution over the period 2000–2020. This product combines data from all available infrared sensors ((A)ATSR, SLSTR, and AVHRR sensors), ensuring good resolution where data are available, unlike other SST products which also include microwave and in-situ measurements, resulting in considerable smoothing of the SST field. Where SST data is not available, spatial interpolation is performed to obtain a cloud free product which, at the cost of resolution on finer features, provides complete synoptic coverage of our large study area. This

interpolation tends to provide an underestimate of the detection of fronts, as the SST field is smoother over cloud-covered areas (Merchant et al., 2019). However, the combination of several sensors allows to reduce these areas to a minimum. Furthermore, we have only considered cloud-free pixels for our analysis, which ensures that cloudy areas are not taken into account in our quantification.

### 2.2   Delimitation of biomes

The three biomes in our domain correspond to well known production regimes which have been described previously, although sometimes with different names. The region is characterized by the presence of a large-scale north-south gradient in Chl-*a*. All pixels where water depth is less than 1500m (red isobath in Fig. 1) are masked to exclude the continental shelf.



The permanently oligotrophic regime to the south of our study area, characterized by warm waters and low Chl-*a* (Fig. 1), is known as the subtropical gyre permanently stratified biome (Sarmiento et al., 2004) or permanent deep Chl-*a* maximum biome

(Bock et al., 2022). Their is no clear physical boundary to the northern limit of this permanent subtropical biome, so we have chosen the latitudinal limit of 32°N to delineate it, which roughly corresponds to the $0.1mgm^{-3}$ Chl-*a* isocontour in annual mean Chl-*a*.

North of 32°N, the seasonal subtropical biome is also mainly oligotrophic, with intermediate levels of Chl-*a* and temperature, and characterized by slightly increased productivity in winter. This second biome is known as the subtropical gyre seasonally

stratified biome (Sarmiento et al., 2004). It is bounded to the north by the meanders of the Gulf Stream jet. The Gulf Stream jet conveys warm, salty waters poleward along the Florida coast up to Cape Hatteras (35°N), where the jet separates from the continental shelf and meanders essentially zonally. The north wall of the jet, so called because of its steep temperature gradient, marks the sharp, sinuous and unsteady northern limit of this biome.

To the north of the Gulf Stream is the Slope sea which extends to the shelf-break, with colder and fresher waters (Linder and

Gawarkiewicz, 1998). Aligned with the shelf break, a persistent front with an intensified surface jet separates the shelf waters (excluded from this study) from the slope sea. This highly productive subpolar biome, known as subpolar waters (Sarmiento et al., 2004) and high-chlorophyll-bloom (Bock et al., 2022), is characterized by a strong spring bloom whose onset is tied to the spring stratification of the mixed-layer.

The position of the north wall of the Gulf stream, that delimits the subpolar and seasonal subtropical biomes (black me-

andering contour in Fig. 1a-b), is determined at each daily time step by thresholding the daily SST map. The daily threshold values (black vertical line in Fig. 1c) are determined from the daily SST distributions above 32°N (blue line, continued by the yellow line in Fig. 1c). Indeed, the Gulf Stream is easily identifiable in this distribution as it is manifested by a temperature peak (yellow line in Fig. 1c). To detect the start of this peak, which marks our boundary, we fit the peak with a Gaussian, and define the threshold temperature as the mean minus twice the standard deviation. Then, the threshold temperature time

series is median filtered over an 8-day window to eliminate spurious detection anomalies. This separation method ensures quasi-unimodal Chl-*a* distributions within each biome (Fig. 1d).

With the above delimitation of biomes, the two atmospherically and topographically fronts, i.e. the Gulf stream and the shelf-break, are located within the seasonal subtropical biome and subpolar biome respectively, while the permanent subtropical biome only contains submesoscale fronts.

## 150 2.3 Front detection

Our front detection method aims to sort the domain into pixels that are located (or not) over fronts, at each daily time step. We follow the approach of Liu and Levine (2016) that builds upon the well established Cayula-Cornillon algorithm (Cayula and Cornillon, 1992; Belkin and O'Reilly, 2009), and identify pixels as belonging to fronts when the region in the vicinity of the pixel is characterized by high SST heterogeneity. More precisely, on each pixel and for each time step, we compute

an Heterogeneity Index (HI) which quantifies the heterogeneity of the SST in a square window comprising a few pixels ($7 \times 7$) centered on the pixel of interest. Thus HI measures the SST heterogeneity at spatial scales less than 30km. The HI

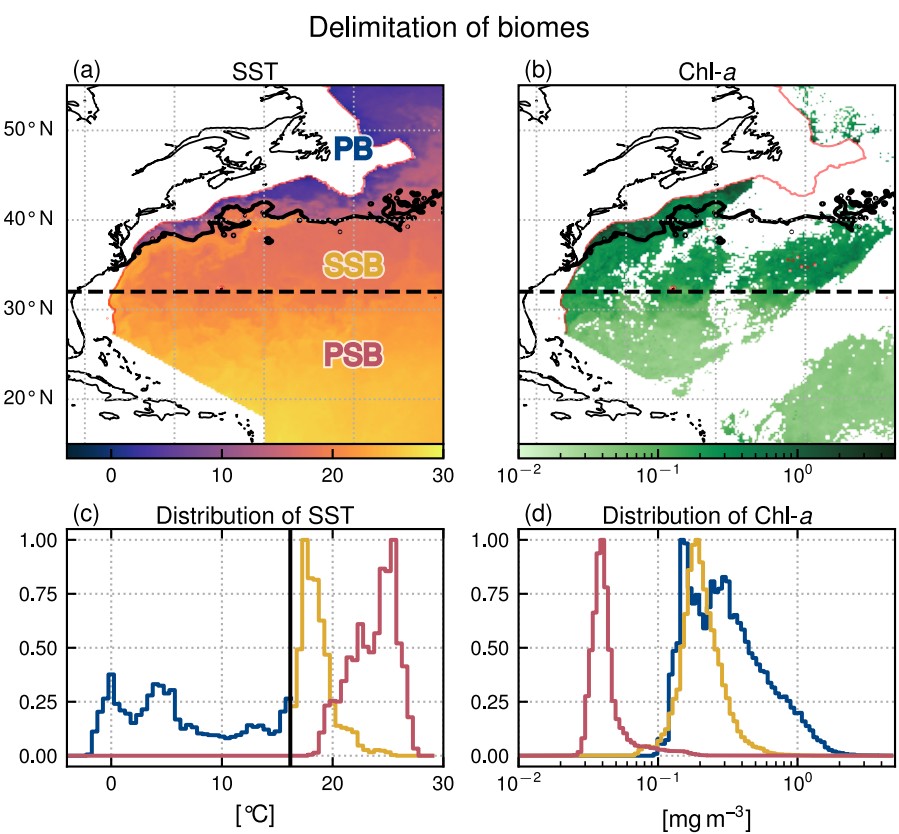

**Figure 1.** Delimitation of the three biomes in the Gulf stream extension region: the Permanent Subtropical Biome (PSB, south of the dashed line at 32°N), the Seasonal Subtropical Biome (SSB, between 32°N and the meandering Gulf stream northern wall on that day marked with the black contour), and the Subpolar Biome (PB, north of the Gulf stream northern wall). (a) SST and (b) Chl-*a* snapshots on the 22 April 2007 (with data masked by clouds in white), (c) SST and (d) Chl-*a* distribution within each biome for the same day (PB:blue, SSB:yellow, PSB:red). The black line in (c) shows the SST threshold value detected to delimit the Gulf Stream northern wall (see methods section). The x-axis of the distributions correspond to the x-axis scale of the corresponding color bars. The red line follows the 1500m isobath. Data on the continental shelf (< 1500m) is not considered here and have been masked.



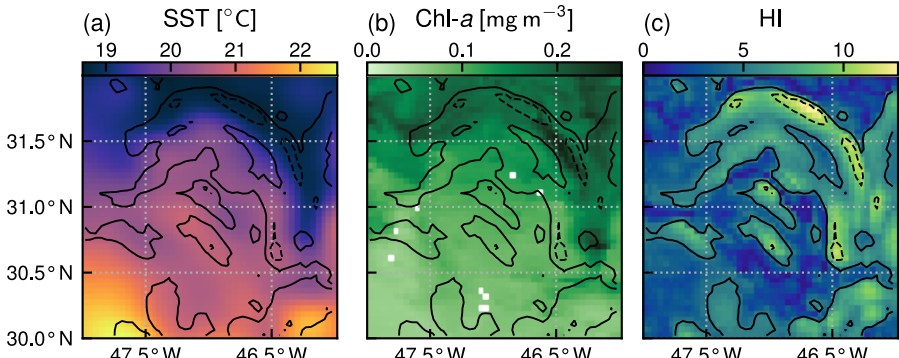

**Figure 2.** The SST, Chl-*a*, and heterogeneity index (HI) of a front on the 7 July 2007. The plain and dashed contours correspond to HI values of 5 and 10. This front is categorized as weak. Chl-*a* are elevated inside the front.

is defined as the weighted sum of the skewness $\gamma$, standard deviation $\sigma$, and bimodality $B$ of SST within the window (HI = $a\,(b\gamma + c\sigma + dB)$, with a, b, c, d constant normalization coefficients).

We have adapted the original formulation of Liu and Levine (2016) with minor modification in the computation of bimodality
and in the relative weighing of each component which are described below. We computed the bimodality as the L2 norm of the difference between the SST histogram (with bins of 0.1°C) and a Gaussian fit of the histogram. We chose this method because it was simpler to implement and more robust than that of Liu and Levine (2016) which computed the normalized absolute difference between a polynomial fit of order 5 of the SST histogram and a Gaussian fit of the histogram, the polynomial fit often being poorly constrained. We normalized each component by its variance (b, c and d being defined as the inverse of
the standard deviation of each component computed over one year), as opposed to Liu and Levine (2016) who normalized the coefficients by their annual maximum. This avoided putting too much weight on extrema. Finally we normalized HI (coefficient a) such that 95% of values are below an arbitrary value of 9.5. For simplicity, the normalization coefficients were computed for year 2007 and used over the entire time series. We performed sensitivity tests on the different parameters used to compute HI, namely the number of pixels in the rolling window and the choice of normalization coefficients. An example of the resulting
HI over a single front is given on Fig. 2; elevated HI values are located along the curved SST gradient to the north east of the window.

At each time step, we sorted the pixels into those belonging to strong fronts (defined as HI > 10), those belonging to weak fronts (defined as $5 < \text{HI} < 10$), and those that do not belong to fronts (when HI < 5, and called them "background" in the following). Sorting fronts by range of HI enabled us to roughly separate the quasi-permanent fronts, which are associated with
the strongest SST gradients (strong fronts), from the more ephemeral ones, associated with weaker gradients (weak fronts). The choice of the two HI threshold is somehow arbitrary, but it is supported by the HI distributions presented below.





## 2.4    Quantification of the impact of fronts on Chl-*a*

We used three metrics to quantify the impact of fronts on Chl-*a*, the excess Chl-*a* E for the local scale (scale of fronts), the surplus Chl-*a* S for the global scale (scale of biomes) and the lag in onset day L for the bloom timing.

In order to quantify how Chl-*a* is affected by fronts at the local scale, the metrics E is based on the comparison between the distributions of Chl-*a* over fronts and the distributions of Chl-*a* outside of fronts. Distributions were computed over 8-day windows to limit the influence of particularly cloudy days. They were computed within each biome, and over latitudinal bands of width 5°, to minimize the influence of the large-scale north south gradient in Chl-*a*. Distributions were compared in terms of their median value, but using mean values yielded similar results (not shown). We defined and computed the local excess Chl-*a*, E (expressed in per cent), as the median value over fronts minus the median value in the background, divided by the median value in the background, for each distribution. We repeated this for weak and strong fronts.

To quantify the large-scale impact of fronts on Chl-*a* at the scale of biomes, we computed the biome surplus of Chl-*a* which we defined and computed as the relative difference (expressed in per cent) between the mean Chl-*a* over the entire biome (MT), and the mean Chl-*a* over the background (MB), $S = (\mathrm{MT} - \mathrm{MB})/\mathrm{MB}$. Thus the surplus S measures the extra quantity of Chl-*a* at the scale of the biome ($\mathrm{MT} - \mathrm{MB}$), relative to what would be the situation in the absence of fronts (MB), and thus accounts for the local increase over fronts (E), but also for the proportion of fronts in a given biome. To better understand the meaning of the surplus, let us consider the simplified case where Chl-*a* is homogeneously doubled over fronts compared to the background value, i.e. when E = 100%; in that case, the surplus Chl-*a* is 50% if there are 50% of fronts, and is 1% if there are 1% of fronts. Note that the computation of the local excess E is based on median values because it relies on the comparison of distributions, while for the computation of the biome surplus S, we used mean values in order to be conservative.

Finally, the time series of the Chl-*a* median in the subpolar biome is characterized by a spring bloom, of which we measured the timing (onset date) both in the fronts and in the background. Because the spring bloom onsets propagates from south to north, these timings were inferred over latitudinal band of width 5°. To extract the onset date, we filtered the Chl-*a* median time series with a low-pass Butterworth filter of order 2 and cutting frequency $1/20\mathrm{days}^{-1}$. The filtered time-series displayed strong variations in their phenology from year to year, but a bloom was always discernible. We considered data from February to July, which allowed us to isolate the spring bloom and exclude the autumn bloom. First, we detected the maximum value of Chl-*a* in this time window, then defined the bloom onset as the time of maximum Chl-*a* derivative prior to the time of maximum Chl-*a*. To estimate the uncertainty in this evaluation, we computed the standard deviation of all days for which the Chl-*a* derivative was above 90% of its maximum value. Finally, we averaged the yearly values of lag L in onset days over the 20 years range by computing a weighted average and standard deviation of the difference between values in fronts and in background for each latitudinal band, with the weights equal to the inverse of the standard deviations for each year.



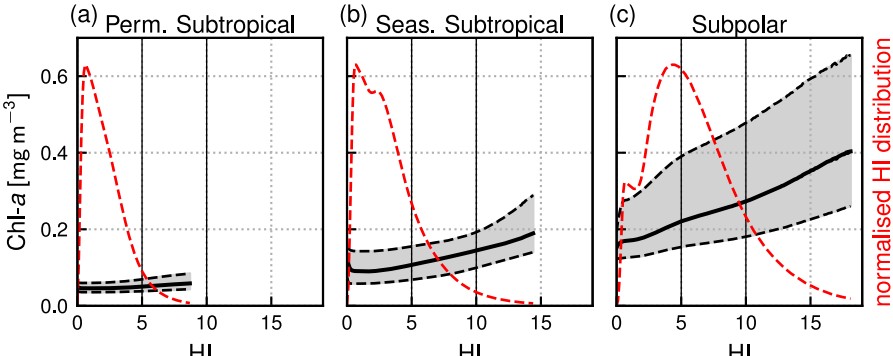

**Figure 3.** Normalized distribution of the Heterogeneity Index (HI, red dashed line) within each biome, and distribution of Chl-*a* as a function of HI (representing front strength), over the full period 2000–2020. Shown are the median value of the Chl-*a* distributions (solid black line), and 1st and 3rd quartiles (dashed lines). Note that 0.5% of pixels have outstanding large HI values and are not included here.

## 3 Results

### 3.1 Distribution of fronts

Our definition of weak fronts and strong fronts is primarily based on thresholds, derived from the HI distributions (Fig. 3, red dashed line). In each biome, the majority of HI values are below 5, and as mentioned before, we used this threshold to distinguish pixels in the background from those over fronts (when HI > 5). Secondly, the number of points in the HI distribution decreases sharply as the HI value increases above 5, reflecting the fact that fronts with stronger SST gradients are much less frequent than fronts with weaker gradients, as one would expect. We used the HI threshold of 10 to distinguish weak fronts from strong fronts. This separation is imperfect, as seen for instance in Fig. 2 where a few pixels with HI values larger than 10 appear at the core of an otherwise weak front.

However, the choice of these two HI thresholds is also guided by the resulting global spatial climatology of weak and strong fronts (Fig. 4). Weak fronts are abundant, and more or less evenly distributed, over a broad band around and north of the Gulf Stream jet (Fig. 4a). To the south of the Gulf Stream jet, weak fronts are less present, with nevertheless more fronts on the edges of the subtropical gyre (around 28°N) than in its center. This distribution of weak fronts is consistent with the predominance of mesoscale variability observed along the Gulf Stream system from satellite altimetry (Zhai et al., 2008), and the injection of eddy kinetic energy north of the Gulf Stream jet by the Gulf Stream extension. It is thus consistent with the generation process of submesoscale fronts through mesoscale strain.

The climatological distribution of strong fronts shows that they coincide mainly with the Western Boundary Current system, which consists of two main permanent fronts, the first following the Gulf Stream jet going northeast from Cape Hatteras, and a second more northerly and extending eastward to 50°W, following the northeastern U.S. continental shelf break (Fig. 4b). Thus these contrasted coverage of localized strong fronts and more widespread weak fronts are consistent with the hypothesis



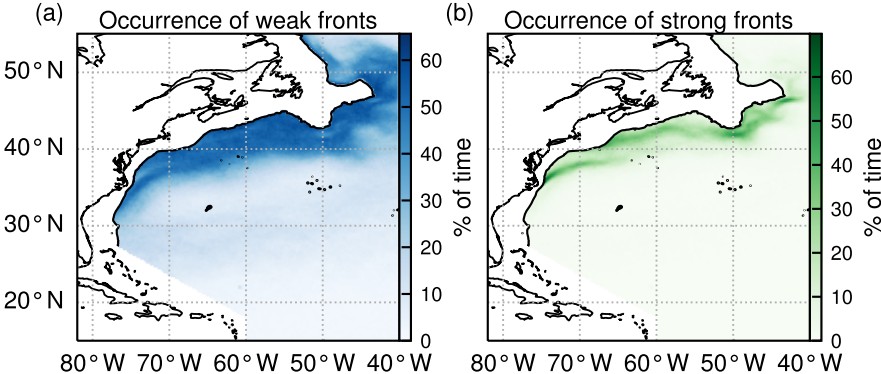

**Figure 4.** occurrence of (a) weak fronts and (b) strong fronts expressed as the percentage of time over the entire time series (2000–2020) that a given pixel is occupied by a front.

that weak fronts are representative of ephemeral submesoscale fronts, whereas strong fronts are representative of the more permanent fronts associated with the Western Boundary Current system that covers part of the seasonal subtropical and subpolar biomes. This separation allows us to provide a first order sorting of the impact of each type of front.

230    With the chosen thresholds, the areal proportion of weak fronts tends to increase from South to North, with 7%, 19% and 42% of HI values comprised between 5 and 10 in the permanent subtropical, seasonal subtropical, and subpolar biome, respectively. Regarding strong fronts, there are only present in the seasonal subtropical and subpolar biome (as expected from the fact that the Gulf Stream and shelf-break fronts are located within these biomes), where HI values above 10 account for 6% and 17%, respectively.

235    The fraction of the area occupied by fronts also varies with the seasons, with generally less fronts in summer (Fig. 5d-f). In the permanent subtropical biome, weak fronts cover on average up to 12% in spring and drop to 2% in summer. In the seasonal subtropical biome, the variation is from 27% to 13% for weak fronts, and 8% to 4% for strong front. In the subpolar biome, strong fronts cover between 11% in summer and 26% at their peak. This seasonality is consistent with other estimates that submesoscale activity is greater in winter due to greater mixed-layer thickness compared to summer (Callies et al., 2015), and also consistent with the most recent modelling results by Dong et al. (2020) in another Western Boundary system (the Kuroshio Extension) where they show that the strongest submesoscales occur with a lag of about a month after the mixed layer thickness maximum is reached. An exception to this seasonal pattern is the greater presence of weak fronts in summer in the subpolar biome (50% in summer versus 35% in winter), which may be partly explained by the reduction of the HI over Western Boundary Current fronts in summer, which results in them being counted as weak fronts in summer.



## 3.2 Local impact of fronts on Chl-*a*

The local impact of fronts on Chl-*a* is illustrated by the example shown in Fig. 2, where the highest values of Chl-*a* are found within the HI contour delimiting fronts. It should be noted, however, that in this example as in other similar examples, the localization of the highest Chl-*a* do not perfectly coincide with elevated values of HI; they are places where the HI is large and Chl-*a* is small, and there are also places with elevated patches of Chl-*a* outside of HI contours.

Nevertheless, it appears that on average, Chl-*a* is affected by the presence of fronts, and depends on their strength. In order to measure that effect, we computed for each biome the Chl-*a* distribution sorted by bins of HI (of width 0.1) for the whole time series (Fig. 3). For low values of the heterogeneity index, these distributions are representative of background conditions and reflect the expected differences between biomes: the median Chl-*a* is lowest ($0.05 mg m^{-3}$) in the permanent subtropical biome, intermediate in the seasonal subtropical biome ($0.1 mg m^{-3}$), and highest in the subpolar biome ($0.2 mg m^{-3}$). The Chl-*a* variability along the year and within the biome (i.e. the width of the distribution, highlighted by grey shading in Fig. 3) is larger moving northwards, because of the higher seasonal variability.

Importantly, in all 3 biomes, Chl-*a* values increase with HI, and are also more dispersed as HI increases (Fig. 3). This suggests that the distinction between background, weak and strong fronts is somehow artificial and that the observed changes in Chl-*a* are overall rather continuously dependent on the value of HI. Nevertheless, our partitioning into background, weak and strong fronts remains meaningful because it enables us to isolate the most permanent fronts from the ephemeral ones, and we will therefore retain it in the following.

Now we examine how the impact of weak and strong fronts varies seasonally in the three biomes, which are characterized by different seasonal variations of Chl-*a* (Fig. 5). We first describe these seasonal variations, then examine the impact of fronts over each biome as a whole, and then by latitudinal bands within each biome.

In the permanent subtropical biome (Fig. 5a), the seasonal variations of Chl-*a* are very modest with a weak peak in winter. In the seasonal subtropical biome (Fig. 5b), the seasonal variations are well marked, with a minimum in summer, and an increase that starts in fall and peaks in late winter. In the subpolar biome (Fig. 5c), there is a marked bloom in spring with a peak in Chl-*a* in April, followed by an autumn bloom albeit with a smaller magnitude. These different phenologies are well documented and largely explained by the differences in the seasonal cycle of the mixed-layer, and the relative depths of the winter mixed-layer and the nutricline (see for instance Lévy et al. (2005) for a description of the drivers of these three production regimes in similar biomes of the Northeast Atlantic). In the permanent subtropical biome, the low productivity is due to the fact that winter mixing is not sufficient to provide a substantial convective supply of nutrients; in contrast, in the seasonal subtropical regime, the increase in production in fall starts when the mixed-layer deepens and reaches the nutricline, leading to a fall-winter bloom. In the subpolar biome, this fall bloom is interrupted in winter when the mixed-layer significantly deepens, diluting cells vertically, and a spring bloom is initiated when the mixed-layer stratifies at the end of winter.

In both subtropical biomes, Chl-*a* over fronts is systematically larger than in the background, and this throughout the year (Fig. 5a-b). This increase is very modest over the permanent subtropical biome. The local increase over weak fronts also remains modest in the seasonal subtropical gyre, but is much larger over strong fronts. Finally in the subpolar biome (Fig. 5c),




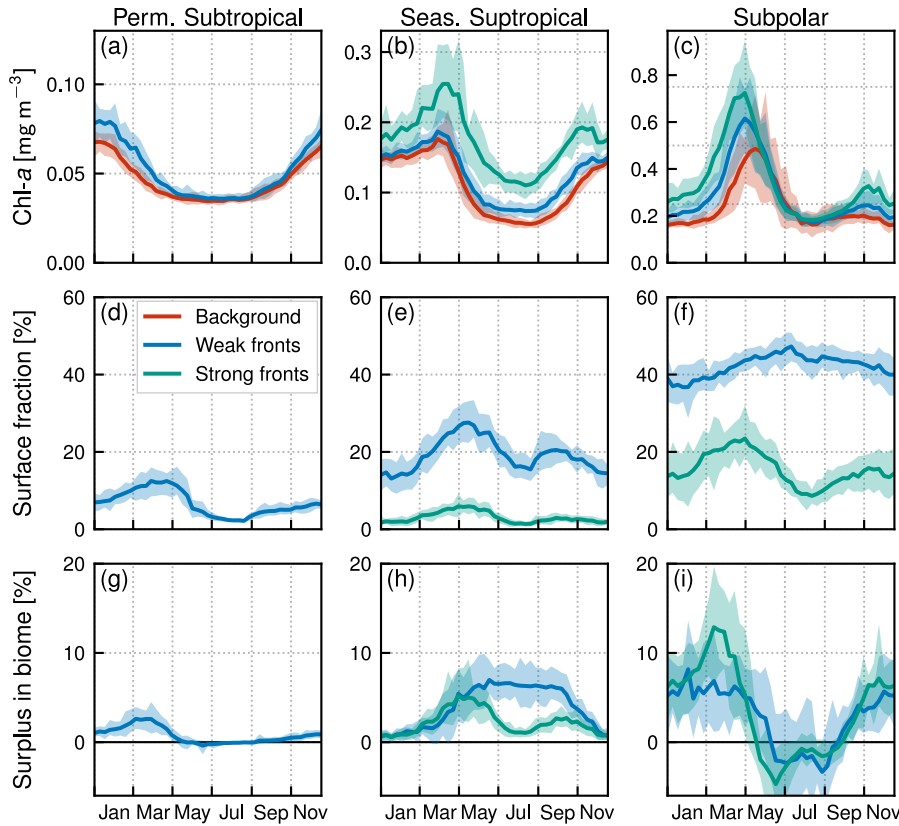

**Figure 5.** Seasonal impact of fronts on Chl-*a* in the permanent subtropical biome (1st column), in the seasonal subtropical biome (2nd column) and in the subpolar biome (3rd column). (a-b-c) Chl-*a* median values (top row) over weak fronts (blue), strong fronts (green) and background (red). The differences between the curves show the local impact at the scale of fronts. (d-e-f) Surface fraction occupied by weak fronts and strong fronts. (g-h-i) Global Chl-*a* surplus due to weak fronts and strong fronts at the scale of the biome. The surplus accounts for the local excess and for the number of fronts (see method). The plain lines represent the climatological mean, and the envelopes mark the standard deviation over the period 2000–2020. Chl-*a* is more strongly enhanced over strong fronts than over weak fronts, but weak fronts are more numerous than strong fronts, resulting in a Chl-*a* surplus that can be reversed.





Chl-*a* is significantly higher over fronts from the fall until the peak of the bloom but this difference diminishes as the spring
bloom decays, and throughout summer. These results are very weakly sensitive to the choice of parameters used to compute HI
(supplementary Fig. A1). We can also note that the standard deviation range of the yearly median values (Fig. 5a-c) is smaller
than the differences between fronts and background, which further confirms that these differences are robust over 20 years of
data, for the three biomes and the two types of fronts.

The strength of the local excess in Chl-*a* over fronts also depends on latitude. In the permanent subtropical biome (Fig. 6),
the excess is only detectable north of 25°N and remains modest (E<10%); it reaches its maximum of 10% at the beginning
and end of the production season. In the seasonal subtropical biome (Fig. 7), the excess is also small in the southern part
of the biome (south of 35°N) but strongly increases further north (between 35°N and 45°N) and decreases again going even
further North (between 40°N and 45°N). Moreover, in this biome, the increase is stronger in summer (July-August) compared
to winter (December-January). Finally in the subpolar biome, the magnitude of the local excess diminishes going northward.
It tends to be stronger during winter and during the bloom, and weaker in summer. In fact in summer, the differences are
hardly discernible when averaged over the entire biome (Fig. 5c), but there is an excess Chl-*a* in the southern part of the biome
(<45°N), and a slightly deficit in the northern part of the biome (>45°N) (Fig. 8).

It ensues that the annual mean local excess in Chl-*a* over fronts has a distinct latitudinal pattern (Fig.9a). The strongest local
increase due to fronts is located in the latitudinal band 35–45°N, where the seasonal subtropical and subpolar biomes meet.
The excess is two to three times larger over fronts North of the Gulf stream (i.e. in the subpolar biome) than South of it (i.e.
in the seasonal subtropical biome). Overall, the annual mean local excess over weak fronts varies between 0 and +30%, and
between 15% and +60% over strong fronts.

### 3.3 Biome-scale impact of fronts on Chl-*a*

Within each biome, the surplus Chl-*a* due to the presence of fronts accounts both for the relative increase of Chl-*a* over fronts
and for the relative area covered by fronts. In the permanent subtropical biome, the surplus due to weak fronts oscillates between
a maximum of 3%, and a minimum of zero in summer when the coverage of fronts is the lowest (Fig. 5g). The magnitude of
the surplus is larger in the seasonal subtropical biome where it reaches 7% in May for weak fronts (Fig. 5h). We can note
that the surplus due to strong fronts is much weaker despite a much stronger local impact of strong fronts (Fig. 5b) due to the
small surface area covered by strong fronts (Fig. 5e). The largest surplus is found in the subpolar biome with maximum values
of 12% and due to strong fronts in March. In contrast to the seasonal subtropical biome, in the subpolar biome the surplus
associated to strong fronts is larger than the surplus associated to weak fronts. We can also note a deficit of Chl-*a* due to fronts
in summer (negative surplus, Fig. 5i) due to the negative excess in the northern part of the subpolar biome (Fig. 8b-d).

Overall, the annual mean Chl-*a* surplus due to fronts is very modest (Fig.9b). All fronts added, the surplus is of the order
of 1% for the permanent subtropical biome and 6% for the seasonal subtropical and subpolar biomes, with contributions from
weak and strong fronts which have similar magnitudes. Therefore, there is roughly 5% more Chl-*a* in the Gulf Stream region
than there would be in the absence of fronts.




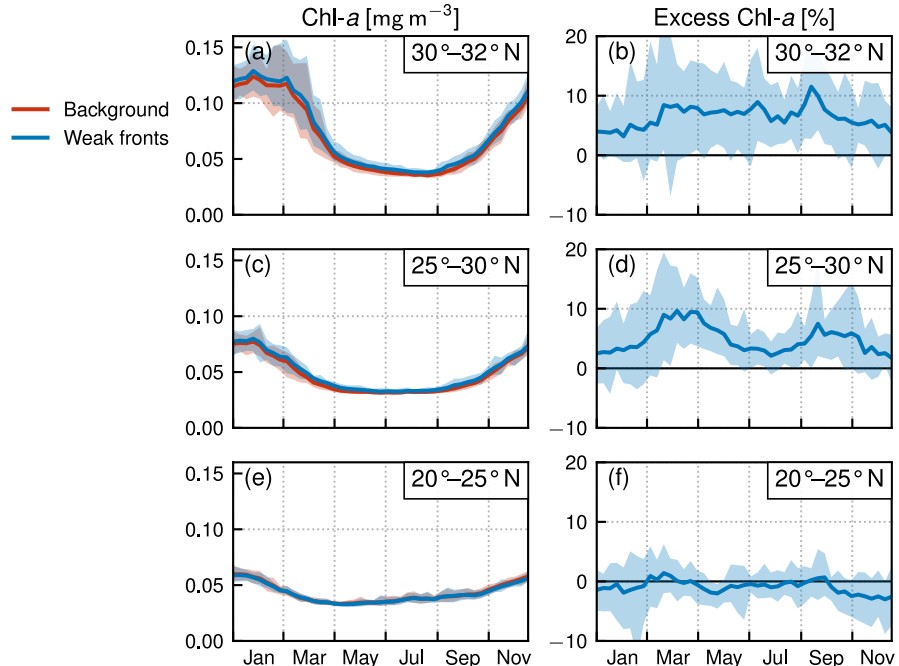

**Figure 6.** Permanent subtropical biome: local impact of fronts on Chl-*a* by range of latitudes in the biome. (a-c-e) Chl-*a* median values over weak fronts (blue) and background (red), (b-d-f) corresponding local excess of Chl-*a* in weak fronts computed as the relative difference of Chl-*a* in fronts and in the background. The plain lines represent the climatological mean, and the envelopes mark the standard deviation over the period 2000–2020. The excess increases from south to north.

### 3.4 Bloom timing in the subpolar biome

Another strong impact of fronts is the earlier onset of the bloom over fronts in the subpolar biome (Fig. 5c). The spring bloom onset propagates from South to North in the biome, starting in early April at 35°N and in late June at 55°N (Fig. 8). The bloom
onsets earlier over fronts in each latitudinal band (Fig. 8). To quantify the lag in bloom onset day over fronts, we estimated bloom onset days for each year and within each latitudinal band, over the background and over fronts (Fig. 10). Our estimates are based on averaged statistics over Eulerian time series, assuming that the bloom evolves coherently in the background (resp. over fronts) within each latitudinal bands in the subpolar biome. Thus we are assuming that front and non front pixels can be pooled apart to follow the bloom evolution over the two contrasting environments. This assumption is suggested by
high-resolution models of the bloom (e.g. Lévy et al., 2005; Karleskind et al., 2011), but it is imperfect as the bloom evolves along Lagrangian trajectories. Nevertheless we found very consistent results with this approach. In all latitudinal bands, we found that the bloom onset occurs one week earlier in weak fronts than in the background (by $-6.7 \pm 1.1$ days) and two weeks earlier in strong fronts (by $-13.5 \pm 1.5$ days). There is a large spread in bloom onset dates in the 20 years of data, due to the very intermittent nature of the bloom onset (Keerthi et al., 2021) which makes it difficult to detect with precision during



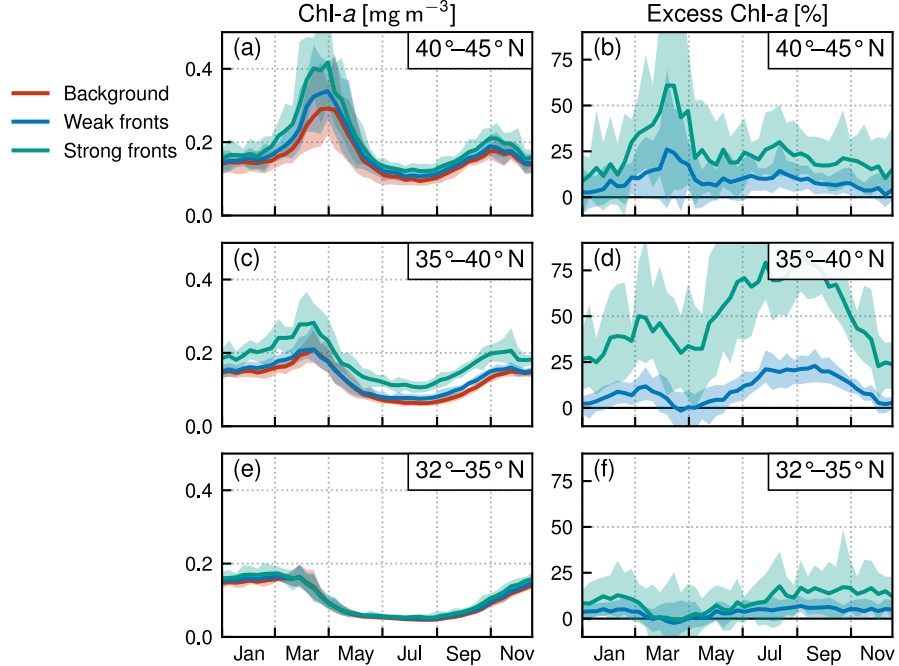

**Figure 7.** Seasonal subtropical biome: local impact of fronts on Chl-*a* by range of latitudes in the biome. (a-c-e) Chl-*a* median values over weak fronts (blue), strong fronts (green) and background (red), (b-d-f) corresponding local excess of Chl-*a* in weak and strong fronts computed as the relative difference of Chl-*a* in fronts and in the background. The plain lines represent the climatological mean, and the envelopes mark the standard deviation over the period 2000–2020. The excess is maximum at mid-latitudes.

certain years. Furthermore, in many cases no difference in bloom onset or duration could be detected between the fronts and the background (dots aligned on the diagonal in Fig. 10). Nevertheless, for individual years, delays larger than one months could occur.

## 4    Discussion

Our analyses of surface satellite data over the Gulf Stream extension region, based on the computation of an heterogeneity

index HI, allowed us to show a substantial local increase of Chl-*a* concentrations over SST fronts compared to background levels, to detect earlier blooms by one to two weeks over fronts, and to quantify that the global effect of fronts on Chl-*a* concentration at the scale of the region was less than 5%. The background levels in this region are very contrasted seasonally and geographically, with a productive and highly seasonal subpolar biome North of the Gulf Stream, a more steady oligotrophic permanent subtropical biome to the South, and an intermediate situation in between the two, where a seasonal subtropical biome

prevails. The main results above hold for these three contrasted biomes, although with different intensities, which also depend on the strength of HI.



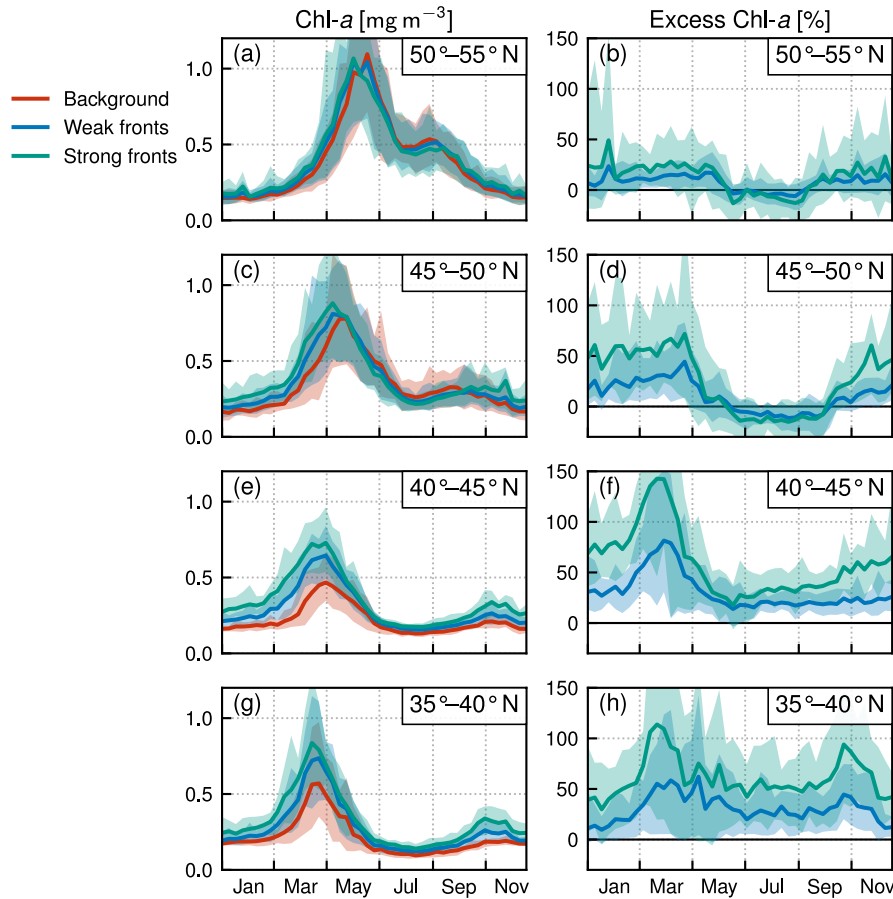

**Figure 8.** Subpolar biome: local impact of fronts on Chl-*a* by range of latitudes in the biome. (a-c-e) Chl-*a* median values over weak fronts (blue), strong fronts (green) and background (red), (b-d-f) corresponding local excess of Chl-*a* in weak and strong fronts computed as the relative difference of Chl-*a* in fronts and in the background. The plain lines represent the climatological mean, and the envelopes mark the standard deviation over the period 2000–2020. The excess decreases from south to north.




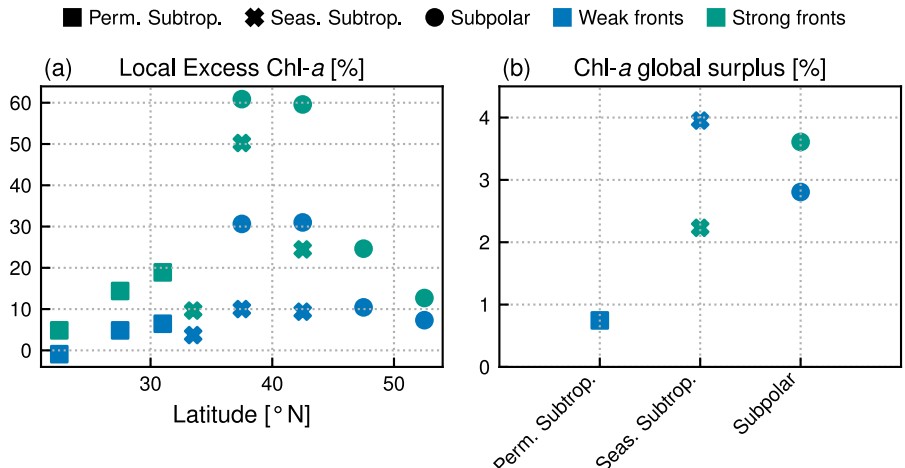

**Figure 9.** (a) Annual mean local Chl-*a* excess over fronts (in %), sorted by latitudinal band (x-axis), by biome (shape of symbol) and by front type (weak fronts in blue, strong fronts in green). (b) Annual mean global surplus of Chl-*a* (in %) for each biome, sorted by front type.

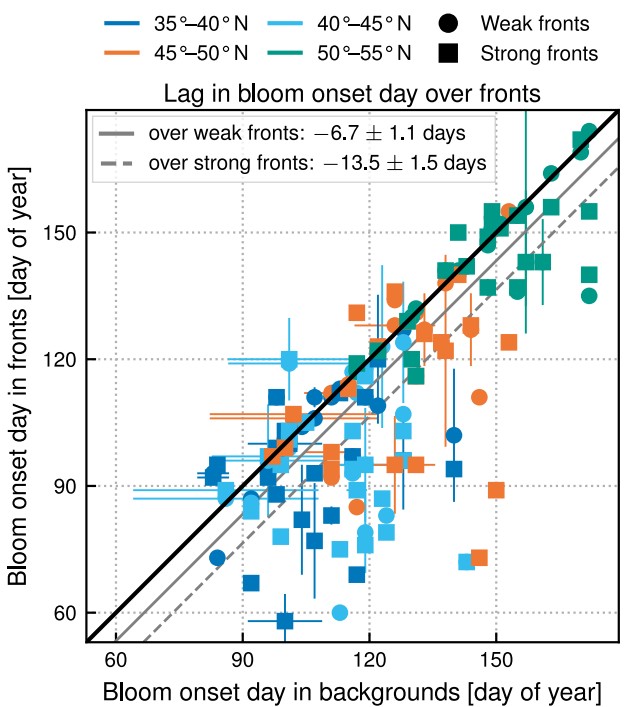

**Figure 10.** Subpolar biome. Comparison of bloom onset dates (in day of year) in the background (x-axis) and over fronts (y-axis), sorted by strength of fronts (shape of symbol), and latitudinal band (color). The line $y = x$ is plotted in black. The distance between the black line and the dotted (resp. dashed) grey line is the measure of the average difference between weak (resp. strong) fronts and background. The bloom onset day propagates from south to north and starts earlier over fronts at all latitudes in the subpolar biome.



## 4.1 Caveats

The level 4 SST product used in this study has the advantage to be readily available on download platforms (here CMEMS) at a reasonably high spatial resolution (4km), avoiding the need to regrid as was done in Liu and Levine (2016) who used
1km resolution L2 MODIS-aqua data. It also has an excellent spatial coverage (e.g. Fig. 1) as it includes merged data from several sensors, but there is a trade-off between coverage and resolution. We have performed initial tests to investigate the differences in using 1km and 4km resolution SST data, which convinced us that the use of 4km resolution data was appropriate (not shown), particularly since the heterogeneity index HI is computed here over boxes of 30km × 30km. Nevertheless a more in-depth study of the sensitivity of our results to the resolution of the satellite products could be carried out in the future.

Another caveat of this level 4 product is that the spatial interpolation performed to merge data from several sensors smooths out the finer features, particularly when some of the data are obstructed by clouds. Here we somehow avoided these smoothed areas by using only cloud-free Chl-*a* pixels. Nevertheless, a bias remains in that there may be a positive correlation between areas with fronts and the presence of clouds. This is the case over the Gulf Stream jet, where dramatic surface temperature gradients are found, and constant clouds are detected over the front. Similar effects can be expected over smaller, short-lived
fronts, but probably on a smaller scale.

Moreover, our assessment of the effect of fronts on phytoplankton, based on surface Chl-*a*, is probably a lower estimate given that the biological signal due to the upwelling of nutrients at fronts often does not reach the surface and is more intense at sub-surface (Mouriño et al., 2004; Ruiz et al., 2019).

Finally, the ratio of Chl-*a* to total phytoplankton biomass in carbon, Chl-*a*:C, changes under varying environmental condi-
tions and community changes (Behrenfeld et al., 2015; Halsey and Jones, 2015; Inomura et al., 2022). Diatoms exhibit higher Chl-*a*:C ratios and are more prevalent in fronts and thus would tend to make our biomass surplus estimation overestimated (Tréguer et al., 2018). This uncertainty could be restricted by taking advantage of recent advances in synoptic estimations of the phytoplanktonic functional types concentrations (El Hourany et al., 2019).

## 4.2 Local impact of fronts on Chl-*a*

With these caveats in mind, we find that the degree of local Chl-*a* increase at fronts varied seasonally, but mostly varied from one biome to another, with an intensity which was weaker in the more oligotrophic region, stronger between 35–45°N, and intermediate in the subpolar biome (Fig. 9a). Moreover, the local excess of Chl-*a* was always significantly larger over strong fronts than over weak fronts. The increase in Chl-*a* with increasing front strength (i.e. with increasing HI, Fig. 3) is consistent with the hypothesis that phytoplankton production is enhanced at fronts by a submesoscale vertical flux of nutrients, and that
this flux is stronger the stronger the front is. On the other hand, the larger dispersion in the Chl-*a* distribution with increasing HI reflects the fact that not all fronts are equally efficient.

Co-occurrence between frontal vertical velocities (or divergence) and enhanced Chl-*a* has been observed over specific fronts in the North Atlantic (Mouriño et al., 2004; Allen et al., 2005; Lehahn et al., 2007). The only study that has statistically connected enhanced Chl-*a* with the presence of temperature front was conducted in the North Pacific subtropical gyre (Liu




and Levine, 2016), which shares characteristics with the permanent subtropical biome examined here. Our results thus extend those of Liu and Levine (2016) to a region with stronger biological contrasts and phenologies.

A key factor determining the magnitude of the local Chl-*a* response to frontal dynamics is the magnitude of the nutrient fluxes, which itself depends on the magnitude of the vertical velocities, of their depth penetration, and of the depth of the nutricline. The nutricline depth shows a sharp latitudinal gradient within this region, from 150m depth at 25°N to 50 m at 375 50°N (Romera-Castillo et al., 2016). This explains the maximum magnitude of the Chl-*a* response at the northern edge of the subtropical gyre, where the lack of nutrients is more severely controlling phytoplankton abundance than further north, and where the nutricline is closer to the surface than further south.

Moreover, vertical velocities associated with ephemeral fronts, often confined to the mixed layer, are likely to be a less efficient nutrient flux pathway to the euphotic zone from the interior than deep, dynamic, persistent fronts extending well 380 below the mixed layer (Lévy et al., 2018). The contrasting impacts of deep and shallow fronts are striking in models (Lévy et al., 2012), but are difficult to quantify from a small number of in situ observations. Here we observed that the magnitude of the Chl-*a* response over fronts increased with the strength of the heterogeneity index HI (Fig. 3). In other words, strong fronts, characterized by high values of HI (HI > 10), led to a stronger increase in Chl-*a* values than weak fronts, characterized by intermediate values of HI (5 < HI < 10).

Finally, an important outcome of this study is that, contrary to what is generally thought, the impact of fronts is stronger in bloom regimes than in oligotrophic regimes. In the subpolar biome, the Chl-*a* excess over fronts reaches 150% during the bloom and 50% in summer (Fig. 8), while in the permanent subtropical biome it never exceeds 10% (Fig. 6). We should note that this last number is lower than that find by Liu and Levine (2016) in the permanent subtropical biome of the North Pacific, possibly because the higher resolution of the product that they have used, as discussed earlier. The enhancement of the spring 390 bloom by submesoscale processes observed here was recently also put forward in a modelling study by Simoes-Sousa et al. (2022).

### 4.3 Permanent and ephemeral fronts

The above results are suggestive that the heterogeneity index could be used as a way to discriminate between permanent (and deep) fronts, and ephemeral (and shallower) fronts. The localization and frequency of strong and weak fronts is consistent 395 with this hypothesis. Weak fronts are much more frequent than strong fronts, as we expect from ephemeral sub-mesoscale fronts compared with permanent fronts (Fig. 4). In addition, the localization of strong fronts coincide with the position of the Gulf Stream. Another element that supports this hypothesis is the scale over which the HI is computed (30km) which gives a strong weight to SST heterogeneities associated with large contrasts which is the case across the Gulf Stream. Of course, more direct evidence linking the penetration of fronts with the intensity of the heterogeneity index would be needed to confirm the 400 association.



## 4.4 Biome-scale impact of fronts on Chl-*a*

The categorization of fronts based on HI has allowed us to quantify the respective contribution of two types of fronts on the regional Chl-*a* budget (Fig. 9b). Weak fronts lead to a local Chl-*a* enhancement which is weaker than strong front, in general, but because they are also more frequent than strong fronts, depending on the biome and seasons, they contribute equally to
the regional Chl-*a* budget as strong fronts. There is also some degree of seasonality in this small surplus of Chl-*a* attributed to fronts, which heavily depends on the region of interest (Fig. 5). As predicted by theory and noted by previous studies, sub-mesoscale fronts — which are confined to the mixed-layer — are less abundant in summer when mixed-layers are shallower. In the south zone, this leads to an overall weaker effect of fronts in summer (near 0%) relative to the rest of the year (less than 3% average), but in the jet area, it is compensated by a larger intensity of the increase in Chl-*a* in summer leading to a
Chl-*a* surplus in summer (7% for weak fronts) which is much larger than in winter (1%). In the north, the situation is quite different with an impact of fronts close to zero during the spring bloom, negative in summer as vertical velocities at fronts are also capable of sinking the surface bloom (Lévy et al., 2018), and maximal in autumn and winter.

Besides these small spatial and temporal variations in amplitude, a key result of this study is that despite strong local impact of fronts, their overall contribution at large-scale remains small, a few percent at most, and of the order of 5% for the entire
region. Nevertheless, this result should be considered as a lower bound, first because increases in Chl-*a* at fronts are often stronger at subsurface than at the surface, and second because in a region characterized by strong gradients like this one, additional nutrient fluxes due to frontal activity might not necessarily lead to local anomalies in Chl-*a*, but could also be hidden by the large-scale gradient.

## 4.5 Earlier blooms over fronts

Another key result of this study is the detection of earlier blooms over fronts than over background conditions in the north of the Gulf Stream jet. Several field and modeling studies have shown that frontal dynamics, by tilting existing horizontal density gradients, increase the vertical stratification of surface mixed layers (Taylor and Ferrari, 2011) and the residence time of phytoplankton in the euphotic zone, leading to early local phytoplankton blooms compared to surrounding areas. However, while the increased stratification over fronts can be directly observed in situ (Karleskind et al., 2011; Mahadevan et al., 2012),
how it affects the timing of the bloom has so far been quantified with numerical models, due to the difficulty in tracking the bloom evolution over fronts which themselves evolve over time (Lévy et al., 2000; Karleskind et al., 2011; Mahadevan et al., 2012). We provide here the first observational evidence of the early onset of blooms over fronts. Moreover, our estimate leads to smaller values (earlier blooms by one to two weeks) than previously estimated from models (20–30 days by Mahadevan et al. (2012)). This effect alone is unlikely to affect productivity budget, but may impact phytoplankton competition at the onset of
the bloom season. The method that we used to quantify differences in bloom timing over fronts and background is based on the time evolution of an eulerian quantity, the Chl-*a* median over latitudinal bands, whereas the bloom evolves along lagrangian trajectories. Considering a rather small area, as we have done here, is a way of overcoming the difficulty of following the temporal evolution on fronts whose life history is too complex to be captured and shorter than the bloom itself. It also limits



the impact that the northward propagation of the bloom could have on the temporal assessment. It should also be noted that
it is inherently difficult to pinpoint the precise onset and end days of a bloom, as the spring bloom shows large intraseasonal
variability in its characteristics; its beginning can be more or less sudden, and is often made of multiple peaks (Keerthi et al.,
2020).

## 5   Conclusions

The Gulf stream extension region is a region of strong biological contrasts and particularly strong frontal activity of the
world's ocean, undergoing rapid warming which strongly affect fisheries (Pershing et al., 2015; Neto et al., 2021). Quantifying
the impact of fronts on phytoplankton there is thus particularly relevant, and we expected to detect a large impact. The use of
20 years of satellite data of SST to detect fronts and of surface Chl-*a* to compute anomalies over the front allowed us to provide
a robust assessment of this impact. We found three main results. First, that the regional increase in surface phytoplankton due
to fronts much weaker than we expected, 5% at most; second that nutrient supplies at fronts enhanced the spring bloom two to
three three times more than they enhanced oligotrophic regions; and third, that the spring bloom onset was earlier over fronts
by one to two weeks, which we already knew from models (Karleskind et al., 2011; Mahadevan et al., 2012) but for which we
had no direct evidence nor sound quantification.

Although limited to the Gulf Stream region, this study provides a well-tested methodology that could enable the study of
the links between small-scale ocean physics and phytoplankton response in other regions of the global ocean. In addition,
these results on the importance of fronts for phytoplankton biomass and phenology could also be used to evaluate models
coupling ocean physics and phytoplankton at high spatial resolution, or to test parameterizations representing the effect of small
scales on phytoplankton production in coarser resolution models. Finally, the combination of these observation-based results
with theoretical arguments and well-assessed models should also allow us to better constrain the response of phytoplankton
production to climate change (Couespel et al., 2021), which still has very large uncertainties as shown by the latest set of Earth
system models (Kwiatkowski et al., 2020).

*Code availability.*   All the scripts needed to reproduce our results, as well as the data necessary to generate the figures in this manuscript are
available at Haëck et al. (2022).

*Author contributions.*   ML, LB and CH conceived the study. CH conceived the methodology and performed the analysis. ML and CH wrote
the paper. All authors contributed to the analysis and discussion of the results.

*Competing interests.*   The authors declare that they have no conflict of interest.





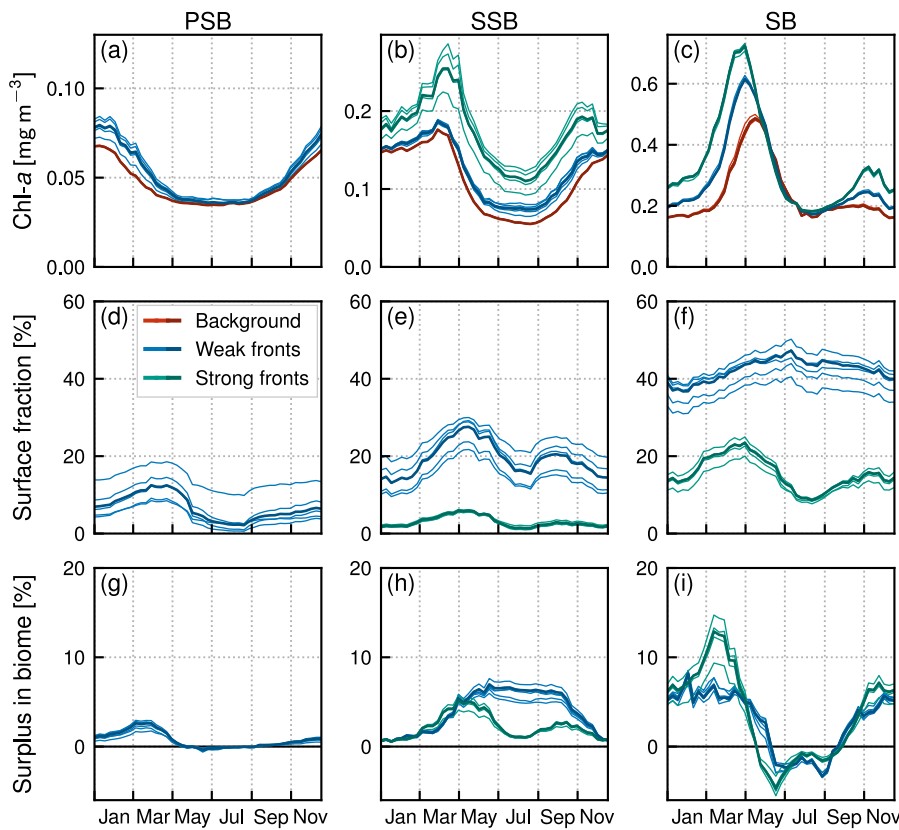

**Figure A1.** Climatological mean of Chl-*a* median values (top row) over weak fronts (blue), strong fronts (green) and background (red), surface fraction occupied by weak fronts and strong fronts (middle row), and global Chl-*a* excess due to weak and strong fronts (bottom row). Each line represent a set of parameter with the bolder line indicating the retained set of parameters. The tested rolling window sizes are 20km, 30km and 40km. Different normalization coefficients are tested for a 30km window size: double the variance, double the bimodality, and double the skewness.

*Acknowledgements.* CH benefited from a PhD scholarship by ENS. The project was supported by TOSCA CNES and by the ENS CHANEL chair. We thank Daniele Iudicone, Francesco d'Ovidio, Sakina-Dorothée Ayata, and Amala Mahadevan for the useful discussions which helped to refine our methodology. We thank Xioa Liu for helping us reproduce their work.

This study has been conducted using E.U. Copernicus Marine Service Information (datasets used: Maritorena et al., 2021; Good et al., 2020b).

GlobColour data (https://globcolour.info) used in this study has been developed, validated, and distributed by ACRI-ST, France.



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
