# Peer review of "Satellite data reveal earlier and stronger phytoplankton blooms over fronts in the Gulf Stream region"

_EGUsphere, 2022_

## Author Response (AR2)

**Dear Editor,**

**Below are our responses to all reviewers' comments, and indications on how we revised our manuscript. Both reviews have been extremely useful and have raised points that we have incorporated in this revision, in particular:**

- **A better explanation of the dynamics associated with persistent and ephemeral fronts**
- **A better justification of the spatial scales at play**
- **An additional supplementary figure with Chl-a distributions**
- **A discussion of the possible effects of fronts on regions that are located outside of fronts, and throughout the text, particular attention to the fact that we are estimated the effect "over fronts" rather than "due to fronts"**
- **A more positive appreciation of our results**
- **The notion that nutrients can be supplied laterally by both types of fronts, particularly with regard to the nutrient stream**
- **The result that phytoplankton can be subducted at fronts**
- **A better description of how the lag in bloom onset date is computed**
- **A shortening of the text when possible**
- **A dozen of additional references**

**After your suggestion, we also have changed the color scheme of some plots and added line markers to help legibility.**

**We thank you and the reviewers for your help,**

**Marina Lévy on behalf of all co-authors**

**RC1:** The manuscript by Haeck et al., presents an analysis of the amplification of surface chlorophyll along weak and strong fronts (detected from satellite SST) in the Western North Atlantic region. The results show significant amplification of chlorophyll at the location of SST fronts, which, when averaged over the domain where fronts are found, increases the total chlorophyll by up to 5% on average over broad biomes. Furthermore, spring blooms are observed to occur earlier — by up to two weeks — along fronts in the subpolar biome.

Heterogeneity in the marine environment is a topic of great interest from physical to biological oceanography. Fronts and eddies usually draw a lot of interest, but the actual impacts on marine ecosystems remain poorly quantified. The manuscript by Haeck et al. thus sheds light on a topic of current relevance.

I found the manuscript very well written and the analysis thorough and sound. This is clearly the result of a lot of work, and the results are not only useful (because they address long standing questions), but also very stimulating. I can imagine a global

extension of the approach, which would further advance our understanding of physical-biogeochemical coupling in the ocean, and inform ecological thinking. I recommend publication of the manuscript with only minor suggestions.

*We thank the reviewer for their positive appreciation of our work*

As a general criticism, I somewhat disagree with the negative spin on the results by the Authors. While 5 % amplification over large scales may not seem that large, it may still have important implications for ecology; after all, chlorophyll amplification can be significant along fronts (E.g. Fig. 8f,h); early onset of blooms along front may also be important for organisms phenology. Furthermore, the study quantifies only one aspect of the effects of front on phytoplankton — i.e., is the direct chlorophyll amplification; as discussed in the text, also because of the ephemeral nature of fronts, nutrients upwelled along fronts may be dispersed more broadly and contribute to an average amplification of chlorophyll that may not be directly co-located with fronts. Of course, this effect is hard, if not impossible to detect purely from remote-sensing. But in few instances in the abstract and conclusions the Authors could be more specific in stating that they quantify the specific effect of biomass amplification at fronts, not other possibly more widespread effects. Additionally, SST is an imperfect proxy for the expected phytoplankton response along fronts, which may lag more than a week relative to the excess of recently upwelled nutrients that may co-occur with cooler SSTs. While these caveats are discussed in the manuscript, they could be somewhat better highlighted.

*We thank the reviewer for their constructive criticism. We agree on all points raised here:*

*1) Regarding the negative spin on the results: in the context of previous observational and modeling studies that have shown very large amplifications at ocean fronts, our initial intention was to put them into a broader context, and to show that these large local amplifications contribute modestly at the regional scale. We agree we might have fallen too much on the negative side, and we added a few modifications in the text to give some more insights on the widespread effects. We have changed the last sentence of the abstract and plain language summary into something more nuanced (see answer to specific comment line 16)*

*2) that 5% amplification of surface Chl might lead to greater effects in terms of ecological implications. We have mentioned two studies to illustrate that point in our discussion (section 4.4):*

*Stock CA, Dunne JP, John JG. 2014. Drivers of trophic amplification of ocean productivity trends in a changing climate. Biogeosciences. 11(24):7125–35*

*Lotze HK, Tittensor DP, Bryndum-Buchholz A, Eddy TD, Cheung WWL, et al. 2019. Global ensemble projections reveal trophic amplification of ocean biomass declines with climate change. Proc. Nat. Ac. Sc. 1:201900194*

*3) that nutrient enrichments due to fronts might lead to phytoplankton biomass enhancement outside of fronts.*

*We agree and this result is actually demonstrated for example in our Figure 2, as the Chl-a filament expands outside the area of elevated HI in the north west part of the domain. We have included a paragraph in the caveat section of the*

*discussion on that specific point. And we are now more cautious in the text of the paper in our wording of what may be caused by fronts versus what may be find over fronts.*

*4) that the mismatch between Chla and HI (seen in figure 2) may be due to the time evolution of the flow, to diffusion, and to the time lag between nutrient supply and Chl increase*

*This aspect is discussed together with 3) now.*

Specific comments:

Line 11, "the global enhancement of Chlorophyll-a due to fronts" maybe clarify as "the global enhancement of Chlorophyll-a along fronts"

*We have rephrased every occurrence of "due to fronts" by "induced at the bioregional scale"*

Line 16, "misleading": I would use more nuance here, and avoid this term.

*We agree and are now more nuanced in our new formulation*

Line 17, "budget": maybe "biomass amplification" or "chlorophyll amplification" would be more appropriate, since technically a budget (which implies some balance of different sources and sinks) has not been evaluated. See also "budget" in line 55 — at least clarify what the term means in this context.

*We agree and have removed the term budget everywhere in the paper*

Line 71: this sentence seems a bit obvious; it could be removed.

*The sentence has been removed*

Line 122, "All pixels where water depth is less than 1500m are masked to exclude the continental shelf.". This needs at least a sentence to justify the removal of shelf waters.

*We agree and we are now more explicit on the fact that our study is focused on the open-ocean*

"Front detection" section. This is well described and builds nicely on previous work. I would only advise to clarify better why some choices were made and how different choices may affect the results; showing or stating that results are not very sensitive to specific thresholds or changes from previous methods would suffice.

*We have brought two paragraphs together to discuss the sensitivity, which is shown in one the supplementary figures*

Lines 204-206: I was somewhat confused by how the lag "L" was defined; maybe add a sentence to clarify its definition.

*We agree and we have entirely rewritten this paragraph for more clarity*

Lines 246-249: This seems an important point; lack of co-location (in space or time) of SST fronts and Chl maxima may be the consequence of interesting dynamical reasons, e.g. related to the timescales of phytoplankton response vs. the physical lifetime of a front. This could be discussed.

*Yes we agree, this relates to one of your earlier comments, and we have added a paragraph in the discussion*

Lines 262-264: this entire paragraph doesn't seem necessary; it could be removed to streamline the paper. In general the paper is on the long side, so some streamlining may help.

*We agree – the paragraph has been removed.*

Lines 306-307: Very interesting to observe the negative effect of fronts in summer, likely related to subduction as discussed later. This is a nice result.

*Thanks. We now highlight this result more clearly in the abstract and in the discussion.*

Lines 315-319, "To quantify …": this part may belong to Methods; it could also be clarified.

*Done*

Lines 351-353: the other effect not quantified here is the broader supply and re-distribution of nutrients that may be caused by fronts; i.e., additional upwelled nutrients may not remain confined to the front that upwelled them, and be able to fertilize phytoplankton more broadly.

*Indeed, we have added one paragraph in the discussion, and a few references.*

Line 385, "what is generally thought": maybe add a reference, or it risks to be a straw man argument.

*Indeed, we have removed this statement and extended the paragraph with more references, including:*

*Oschlies A. 2002. Can eddies make ocean deserts bloom. Glob. Biogeochem. Cyc. 16(4):1106*

*Gruber N, Lachkar Z, Frenzel H, Marchesiello P, Münnich M, et al. 2011. Eddy-induced reduction of biological production in eastern boundary upwelling systems. Nature Geoscience. 4(11):787–92*

*Lathuiliere C, Lévy M, Echevin V. 2010. Impact of eddy-driven vertical fluxes on phytoplankton abundance in the euphotic layer. J. Plankton Res. 33:827–31*

Also, related to the greater impact of fronts in bloom regions relative to oligotrophic regimes, the study by Yamamoto et al. (2018, Journal of Geophysical Research) provided (based on a mesoscale-eddy permitting model) evidence that the majority of nutrient supply to the euphotic zone of the oligotrophic gyres occurs by lateral eddy fluxes rather than vertical eddy fluxes. This certainly relates to the argument of deep nutriclines in these regions.

*Yes, we agree and this is a very important point. We thank you and the second reviewer for raising it. We have changed our introduction, discussion and conclusion to highlight that increased Chl-a over fronts is not necessarily the result of vertical advection but may also be related to lateral transport and particularly to the nutrient stream (as our results actually show evidence for it, and we saw it thanks to your feedback). Added references include:*

*Long Y, Guo X, Zhu X-H, Li Z. 2022. Nutrient streams in the North Pacific. Prog. Oceanogr. 202:102756*

*Pelegrí JL, Csanady GT, Martins A. 1996. The North Atlantic nutrient stream. Journal of oceanography. 52(3):275–99*

*Yamamoto A, Palter JB, Dufour CO, Griffies SM, Bianchi D, et al. 2018. Roles of the Ocean Mesoscale in the Horizontal Supply of Mass, Heat, Carbon, and Nutrients to the Northern Hemisphere Subtropical Gyres. J. Geophys. Res. Ocean. 123(10):7016–36*

*Spingys CP, Williams RG, Tuerena RE, Garabato AN, Vic C, et al. 2021. Observations of Nutrient Supply by Mesoscale Eddy Stirring and Small-Scale Turbulence in the Oligotrophic North Atlantic. Global Biogeochemical Cycles. 35(12):e2021GB007200*

Lines 420-424: a clear sense of why blooms occur early along fronts is a bit hidden in this explanation, maybe clarify a bit.

*We have reformulated the sentence*

Line 429: I think the results of early blooms on fronts may be important for phenology of zooplankton, and potential their predators, which could be more explicitly discussed.

*You are right and we have removed the sentence stating that early blooms do not have strong implications.*

Line 444, "due to fronts": again here, it may be better to add more nuance, since only a direct effect of fronts on co-located chlorophyll was quantified here; indirect effects (e.g. nutrient redistribution) may be also important.

*Yes, we propose to use "associated with fronts" instead of "due to fronts"*

Technical comments:

Line 14: "the the" —> "the"Line 64, "spacial" —> "spatial"Line 70, "built" —> "build"Line 82, "contrasted" —> "contrasting"Line 115: "tends to provide an underestimate of" —> "tends to underestimate"Line 139: "Gulf stream" —> "Gulf Stream"Line 276: "and this throughout" —> "and this holds throughout"Line 326, "one months" —> "one month" Caption of fig. 10, and other parts of the text: the use of "rest." May not be completely clear, maybe use the full word (respectively?)Line 429, "budget" —> "budgets"

**All these corrections have been taken into account in the revised manuscript.**

*Thanks again for your time in providing these very useful and detailed comments*

**RC2: This is the review of "Satellite data reveal earlier and stronger phytoplankton blooms over fronts in the Gulf Stream region", by Haëck et al.**

The authors build on the methods from Levine et al. 2016 to investigate the impact of fronts—as diagnosed by a heterogeneity index (HI) based on sea surface temperature—on surface chlorophyll-a—as estimated by remote sensing—in different biomes in the Western North Atlantic. Namely, a permanently subtropical, a seasonally subtropical and a subpolar biome. Based on the HI, they estimate the occurrence of weak (5<HI<10) and strong (HI>10) fronts in each biome. Regions with HI less than 5 were considered to be background. The authors noted that high HI values were associated with quasi-permanent fronts while the weak fronts were more ephemeral. Data resolution (4 km) allows the authors to estimate HI in a square of around 24 km (49 grid points).

The authors find that fronts increase surface phytoplankton by about 5% over the region, which they claim is a "much weaker than expected" impact ( although the authors are not explicit about the sources of the high expectations). Secondly, the authors claim that the nutrient supply at fronts more than doubled chlorophyll concentrations during the spring bloom. Thirdly, the authors claim that the spring bloom over fronts occur one to two weeks earlier than in background regions.

I find the study interesting. It is well written, well organized and the methodology is sound. I do, however, have a few comments which I believe should be clarified before publication.

Although it is not explicitly stated, the hypothesis the authors are testing is: SST fronts impact surface ocean chlorophyll via submesoscale vertical advection due to unbalanced motions. By stating that the the impact of SST fronts is "much weaker than expected", they seem to reject that hypothesis.

The premise is that the impact of SST fronts on surface chlorophyll occurs via submesoscale motions (e.g. ageostrophic secondary circulations due to frontogenesis). By supplying nutrients into a nutrient-depleted surface layer via submesoscale vertical advection, biomass would increase. The problem with this premise is that it is only valid in the permanently oligotrophic region. In the seasonally subtropical and the subpolar biomes, where fronts are more frequent and stronger, lateral advection plays a much more important role due to large-scale climatological horizontal gradients in surface properties. In the Gulf Stream region, not only is the current know to be a "nutrient stream" (Pelegri and Csanady, 1991), with robust horizontal transport of nutrients, but also the front is maintained by the thermal wind shear. There certainly is imbalance and ageostrophic motion, but I am not sure if that is truly discernible from the submesoscale instabilities the authors seem to be mostly concerned about.

***We are very grateful to the reviewer for their positive assessment of our work, and especially for their sound criticism that we should have examined and discussed the role of lateral advection more thoroughly. This was indeed missing and is an important part of the story. We agree that particularly in the two biomes that are more affected by the Gulf Stream, it is very likely that the signal that we detect is due to the lateral supply of nutrients by the Gulf Stream. This view can explain why the signal is maximum and so much stronger than elsewhere at around 40°N in our figure 9a. We have added this in our discussion and highlighted it in our results and abstract (see also our answer to referee 1 on this same point).***

In addition, it is unclear whether or not the impact of mesoscale eddies on chlorophyll is removed from the analysis by considering a 24 km square. It is known that the curvature induced by mesoscale eddies impact chlorophyll-a distribution (Siegel et al. 2011). While the analysis was performed on a larger scale in the case of Siegel et al. 2011, it is unclear if the mesoscale signal is removed from the current analysis. Could, for example, some of the detected frontal regions be associated with the edges of mesoscale eddies?

*We believe that considering a 24 km square removes a large part of the mesoscale signal from the analysis since mesoscale eddies in this region have diameters roughly between 50 and 200 km. But indeed, mesoscale and submesoscale dynamics are often strongly intertwined, and eddy edges can be considered as small frontal areas and can behave as such. In fact our analysis is better described in terms of frontal dynamics than mesoscale or submesoscale. We have extended on the specific dynamics of persistent and ephemeral fronts, and use the wording front in place of submesoscale, most of the time. We also added a paragraph to better describe the scales that we are examining.*

With regards to the permanently stratified, there's observational evidence (Johnson et al. 2010) that episodic injections, due to meso- or submesoscale processes induce vertical advection of nutrients but that these nutrients get consumed before reaching the surface, so the signal may not be captured by SST in these regions. In addition, there may be photo-inhibition which prevents phytoplankton to be near the surface in oligotrophic regions, even though they may be impacted by frontal motions. The authors partially mention that in lines 351-353, but the studies they cite make use of the QG-omega equation to estimate the vertical velocities, which may be on the mesoscale end of the frontal structures.

*Indeed, we have added reference to Johnson et al and now also mention photo-inhibition. The studies we refer to were cited because they showed nice subsurface Chl-a signals associated with fronts.*

Lastly, it is not clear how the authors reconcile the "considerable evidence that submesoscale motions influence nutrient and light environments" with the "smaller than expected" impact of fronts on chlorophyll-a. In addition, conclusions from this work seem to differ from those of Lie and Levine 2016, which showed a 40% increase in the winter. Is it only due to the different resolution in the SST and chlorophyll-a products?

*We have removed "smaller than expected". What we intended to say here, is that the impact of the regional scale are smaller than the impact at the scale of the front. We have now rephrased this sentence.*

*We are also more cautious in our revision, and this answers some of the concerns of the first reviewer as well, on the fact that the "small impact" may in fact 1) be underestimated and 2) have stronger impacts on the ecosystem as a whole.*

*Also, indeed, the impact of 40% find by Liu and Levine is larger than our findings in the permanent subtropical biome. And it is a very good question to ask why. It could be indeed be due to the resolution of the products, to the size of the window*

*to compute the HI index (10 km for Liu and Levine versus 30 km here), or to the difference between the two regions. Thus we would need to compare the two products in the two regions (Atlantic versus Pacific) for a range of window sizes to understand what drives the difference. We did some tests with their product, but only on a few images, and that did not seem to play a significant role. We also compared the results for different window sizes (see Supplementary Fig 1): in the permanent subtropical biome (PSB) in winter, there is indeed some degree of sensitivity, larger than elsewhere, mainly because the Chl-a values are very small, and the median value is very sensitive there to the area covered by fronts. But a more in depth study comparison, based on a statistically significant number of cases would be required. This would go beyond our objectives here but we intend to test it in the future, as we intend to extend our approach to larger zones of the global ocean. We added a paragraph in our discussion to emphasize the difference and the possible causes.*

Specific comments:

Title: "earlier and stronger" than what?

*The title says "earlier and stronger phytoplankton blooms over fronts" so implicitly, "than outside of fronts". We felt it would be a bit heavy to add this precision to the title, and we have hence explained it more clearly the abstract.*

Lines 29-31: "largely explained by consistent physical forcing and environmental conditions" - not clear what the authors are referring to. Aren't the environmental conditions a consequence of the physical forcing?

*Not necessarily, if one thinks in terms of the nutrient distributions for instance, they are related to the physical forcings but also to their sources and sinks. We have made it clear that by environmental conditions, we were thinking of nutrients.*

Line 31 : "wind-driven circulation": the authors could be more explicit in saying that negative wind stress curl induces downwelling in the subtropical gyres while the opposite occurs on subplolar gyres.

*Done*

LIne 149: the authors seem to differentiate the "atmospherically and topographically"-controlled fronts from "submesoscale" fronts. This distinction is not so straightforward as submesoscale

*This differentiation comes from the fact that there are indeed various ways to define submesoscale, either in terms of a specific scale range or in terms of Rossby number. Here we think of submesoscale as flows characterized by Rossby number of order one, and these flows encompass what we call "ephemeral submesoscale fronts", and "persistent fronts" (such as the Gulf Stream here), which also have Rossby number of order one but have larger scale. We have changed the corresponding paragraph in the introduction to reflect on this.*

Figure 2c: I suggest a qualitative or discreet colorbar for the different categories of HI. In other words, what matters to the reader are the intervals, not the continuous values.

*In Fig 2c, we have overlaid the contours for HI=5 and HI=10 to the color bar. The same contours are overlaid in Fig 2a and 2b. This way, we both show the intervals, and we also show the continuous values. Indeed, part of our discussion is to say that there is a continuity in the relationship between HI and Chl, as shown in Figure 3.*

Line 197: typo. "Onsets propagates". ~Remove "s".

*thanks*

LIne 221: It is also consistent with baroclinic instability, which is not a submesoscale process.

*This has been revised*

Line 237: typo : add "s" to "front".

*thanks*

Lines 243-244: not clear. Are you saying that strong fronts are counted as weak fronts during summer? Isn't the methodology robust enough to have a consistent classification?

*Indeed this was not clear, all the more that there is in fact also a weak reduction in the number of fronts in the subpolar biome in summer. We have removed this comment which was unclear and not supported by further analysis.*

Line 249: I assume this does not affect the interpretation of regions affected by fronts right? Otherwise, it would required a more detailed analysis of points with high chlorophyll-a in regions of low HI.

*Please see our answer to the first reviewer on that specific point. As a more detailed analysis, we have added one figure in supplementary material which shows the distributions of Chl-a (in addition to their median value) within each region and for each season. This figure shows that the quantity of high Chl-a points in regions of low HI (i.e. in the background) is much less than in the regions of large HI (the fronts), and that the median value captures that.*

References

Johnson, K., Riser, S. & Karl, D. Nitrate supply from deep to near-surface waters of the North Pacific subtropical gyre. *Nature* **465**, 1062–1065 (2010). https://doi.org/10.1038/nature09170

Pelegrí, J. L., and Csanady, G. T. (1991), Nutrient transport and mixing in the Gulf Stream, *J. Geophys. Res.*, 96( C2), 2577– 2583, doi:10.1029/90JC02535.

Siegel, D. A., Peterson, P., McGillicuddy, D. J., Maritorena, S., and Nelson, N. B. (2011), Bio-optical footprints created by mesoscale eddies in the Sargasso Sea, *Geophys. Res. Lett.*, 38, L13608, doi:10.1029/2011GL047660.

[Figure]

New supplementary figure showing the distributions of Chl-a by season, biome, and front type.